# Co-Surveillance of Rotaviruses in Humans and Domestic Animals in Central Uganda Reveals Circulation of Wide Genotype Diversity in the Animals

**DOI:** 10.3390/v15030738

**Published:** 2023-03-13

**Authors:** Josephine Bwogi, Charles Karamagi, Denis Karuhize Byarugaba, Phionah Tushabe, Sarah Kiguli, Prossy Namuwulya, Samuel S. Malamba, Khuzwayo C. Jere, Ulrich Desselberger, Miren Iturriza-Gomara

**Affiliations:** 1EPI Laboratory, Uganda Virus Research Institute, 51–59 Nakiwogo Road, Entebbe P.O. Box 49, Uganda; 2Department of Paediatrics and Child Health, College of Health Sciences, Makerere University, Kampala P.O. Box 7062, Uganda; 3Department of Microbiology, College of Veterinary Medicine and Biosecurity, Makerere University, Kampala P.O. Box 7062, Uganda; 4Northern Uganda Program on Health Sciences, c/o Uganda Virus Research Institute, 51–59 Nakiwogo Road, Entebbe P.O. Box 49, Uganda; 5Institute of Infection, Veterinary and Ecological Sciences, University of Liverpool, Liverpool L69 3BX, UK; 6Malawi Liverpool Wellcome Research Programme (MLW), Blantyre P.O. Box 30096, Malawi; 7Department of Medical Laboratory Sciences, Faculty of Biomedical Sciences and Health Profession, Kamuzu University of Health Sciences, Blantyre P.O. Box 30184, Malawi; 8Department of Medicine, University of Cambridge, Cambridge CB2 1TN, UK

**Keywords:** rotavirus, domestic animals, co-surveillance, genotyping, epidemiology, Uganda, inter-host species transmission

## Abstract

Rotavirus genotypes are species specific. However, interspecies transmission is reported to result in the emergence of new genotypes. A cross-sectional study of 242 households with 281 cattle, 418 goats, 438 pigs, and 258 humans in Uganda was undertaken between 2013 and 2014. The study aimed to determine the prevalence and genotypes of rotaviruses across co-habiting host species, as well as potential cross-species transmission. Rotavirus infection in humans and animals was determined using NSP3 targeted RT-PCR and ProSpecT Rotavirus ELISA tests, respectively. Genotyping of rotavirus-positive samples was by G- and P-genotype specific primers in nested RT-PCR assays while genotyping of VP4 and VP7 proteins for the non-typeable human positive sample was done by Sanger sequencing. Mixed effect logistic regression was used to determine the factors associated with rotavirus infection in animals. The prevalence of rotavirus was 4.1% (95% CI: 3.0–5.5%) among the domestic animals and 0.8% (95% CI: 0.4–1.5%) in humans. The genotypes in human samples were G9P[8] and P[4]. In animals, six G-genotypes, G3(2.5%), G8(10%), G9(10%), G11(26.8%), G10(35%), and G12(42.5%), and nine P-genotypes, P[1](2.4%), P[4](4.9%), P[5](7.3%), P[6](14.6%), P[7](7.3%), P[8](9.8%), P[9](9.8%), P[10](12.2%), and P[11](17.1%), were identified. Animals aged 2 to 18 months were less likely to have rotavirus infection in comparison with animals below 2 months of age. No inter-host species transmission was identified.

## 1. Introduction

Species A rotaviruses (RVAs) are a major cause of diarrhoea in children. Despite a significant reduction in rotavirus mortality since the introduction of rotavirus vaccines in many parts of the world, in 2017–2018, 208,009 (95% CI; 169,561 to 259,216) deaths were estimated to occur globally among children under the age of five years old [1]. In central Uganda, 37% of children under five who were admitted to four hospitals in central Uganda with diarrhoea in 2012 to 2013 had rotavirus infection [2], but no deaths were reported following these admissions. Rotavirus diarrhoea, if not well managed, causes high mortality in children and decreases quality of life [3,4]. 

RVA causes diarrhoea in the young of domestic animals, including pigs, cattle, and goats [5]. The prevalence of RVA in animals varies by species and age [5]. RVA infection in domestic animals causes economic losses through costs in management of the disease, decreased production, and mortality of the affected animal [6,7,8,9]: A study in Denmark found the piglets infected with RV weighed 0.5 kg less at 30 days old compared with those that were not infected [6].

RVA disease is mainly prevented by vaccination. Currently, six vaccines have been licensed for use in humans: Rotarix™ (GlaxoSmithKline, Belgium), which contains G1P[8]; RotaTeq™ (Merck and Co, USA), which contains a mixture of four mono-reassortants carrying the genes encoding human G1–G4 and P[7] from the bovine strain and the fifth reassortant strain containing P1A[8] genotypes from a human strain; Rotavac™ (Bharat Biotech International Ltd., India), a bovine–human reassortant contains G9P[11]; Lanzhou lamb rotavirus vaccine (LLR) (Lanzhou Institute of Biological Products, China), which contains G10P[12]; Rotavin-M1™ (POLYVAC-Vietnam, Vietnam), which contains G1P[8]; and RotaSiil™ (Serum Institute of India Ltd., India), which is a lyophilized pentavalent vaccine containing human–bovine reassortant strains G1, G2, G3, G4, and G9 [10]. The available vaccines for use in bovines are a bivalent vaccine containing G10P[11] and G6P[1], and a monovalent vaccine containing G6P[5] [11]. Monovalent and multivalent rotavirus vaccines have been developed for humans and domestic animals, with the aim of providing sufficient cross protection either by incorporating a variety of the most frequent or dominant genotype circulating globally. However, the rotavirus genotypes associated with infections in humans and animals have been fluctuating over time, thus justifying the need for continued rotavirus surveillance in humans and animals [12,13,14]. 

At the time of this study, there was no routine rotavirus vaccination in humans and animals in Uganda. Rotarix™ (GlaxoSmithKline) is available in a few private facilities and was introduced into the routine immunization program in June 2018. The current Rotavirus vaccination coverage is 81% for two doses, as per the Uganda DHIS2 report. To date, there is no animal rotavirus vaccination program in Uganda.

Globally, the most common G- and P- genotypes of RVAs circulating in humans are G1P[8], G2P[4], G3P[8], G4P[8], G9P[8], and G12P[8] [15]. In animals, the G- and P-genotypes of RVAs varies by animal species [14,16,17,18]. The most common porcine RVA genotypes are G3–G5, G9, and G11 in combination with P[6] and P[7] [18,19]. The most common bovine RVA genotypes are G6, G8 and G10 in combination with P[1], P[5] and P[11] [18]. The most common RVA genotypes in felines are G3P[3] and G3P[9], while in canines its G3P[3] [19,20,21].

Despite the importance of domestic animals for household economies in Africa [22], we found no reports on the RVA burden and genotype distribution in cattle and goats, and limited data on porcine rotaviruses in East Africa [23]. This study aimed to assess the prevalence and genotype distribution of rotaviruses in cattle, goats, and pigs, as well as in humans living in the same household in order to determine the potential for the cross-species transmission of rotaviruses and contribute to the available data in Africa. 

## 2. Materials and Methods

### 2.1. Study Design and Population

The study was carried out from December 2013 to January 2014. This was a cross-sectional study of domestic animals (cattle, pigs, and goats) and humans in the community. Cattle, pigs, and goats are the most common domestic animals kept in Uganda, and thus were chosen to be studied. The Masaka District, where the study was carried out, is located in central Uganda and had a record high number of pigs (236,150) in the 2008 country census [24]. The district also had 224,600 cattle and 244,706 goats. The human population of Masaka was projected to be 242,200 in 2011, with an average household size of 4.3 persons. 

The samples sizes for the animals studied were calculated using the Kish and Leslie formula [25] and were estimated to be 323 for cattle, 369 for pigs, and 384 for goats using previously observed rotavirus infection prevalence in cattle with 30% [26] and pigs with 40% [27]. No rotavirus prevalence studies were found for goats; thus, a prevalence of 50% was used in the calculation of the sample size.

Similarly, as we did not have data on the possible prevalence of rotavirus infection and/or diarrhoea for humans in the community, a prevalence of 50% was assumed. Using the Kish and Leslie formula [25], the minimum sample size for the study of humans was estimated to be 384.

### 2.2. Study Participants

Animals of all age groups were studied, but younger animals (8 weeks and below), when present, were selected as previous reports have indicated that younger animals have a higher prevalence of rotavirus diarrhoea compared with older animals [22]. Animals, with and without diarrhoea in the two weeks preceding recruitment, were sampled. The animals were consecutively recruited, from 2–4 parishes (purposively selected), per sub-county in the Bukoto and Buddu Counties, Masaka District, Uganda (Appendix A). A maximum of five animals per household per species were recruited. When more than five animals of one species were found in a home, five animals of that species were randomly sampled.

Children and adults, with and without diarrhoea symptoms in the previous three weeks, staying in the homes where the animals were recruited from, were included in the study. A maximum of five people, randomly selected, per household were recruited. 

### 2.3. Data and Sample Collection

A structured questionnaire was used to collect data on the humans, animals, and households. The social demographic data collected on the recruited humans included information on hand hygiene, water source, and toilet use. Data collected on animals included the age and sex of animal, area of residence, whether the animal was suckling or not, and history of diarrhoea in the previous two weeks. Approximately 5 mL or 5 g of stool was collected from the rectum of the animals using gloved hands smeared with KY Jelly (Johnson and Johnson, Sante Beaute’ France SAS, Sezanne France). The human participants provided 5 g of formed stool or 5 mL of semi-formed stool for the investigation of rotavirus infection.

### 2.4. Laboratory Investigations

Laboratory tests were carried out at the Expanded Programme on Immunisation Laboratory (EPI) at the Uganda Virus Research Institute. Screening for the rotavirus VP6 antigen in animal stool specimens was performed using the ProSpecT Rotavirus ELISA kit (Oxoid Ltd., Hampshire, UK), following the manufacturer’s instructions. Screening for rotaviruses in human samples was performed using NSP3 targeted real time RT-PCR, which is more sensitive than RV ELISA for detecting rotaviruses [28]. 

A subset of rotavirus ELISA negative samples (60 samples) from animals aged 2 months and below was also screened using the NSP3 real-time RT-PCR for quality control of ELISA testing. Samples from younger animals were tested because these were more likely to be positive for rotavirus, as per previous reports [29]. 

Rotavirus dsRNA was extracted from 10% faecal suspensions (200 µL of liquid sample or a bacteriological loop full (the size of a garden pea) from semi-solid samples to 2 mL PBS) using the guanidinium isothiocyanate silica method [30]. The RNA was transcribed to cDNA using M-MLV reverse transcriptase and random hexanucleotide primers (Invitrogen, Life Technologies, Carlsbad, CA, USA) [31], and the resulting cDNA was used in the NSP3 real time RT-PCR and the subsequent VP7 and VP4 PCRs [28,31]. 

All of the samples that were tested to be RV positive on the RV ELISA or the NSP3 real time RT-PCR were genotyped using G- and P-genotype specific primers in nested RT- PCRs, as described previously [31,32,33,34,35]. The positive animal samples were genotyped using methods designed primarily for animal rotaviruses and for human rotaviruses to capture the maximum possible strain diversity, while the positive human samples were typed with methods for human rotavirus genotyping [31]. One human sample that tested positive could not be assigned a genotype using the nested RT-PCR method [31,32,33,34,35]. This sample had adequate viral load, as determined by the VP6-specific qPCR, thus its VP4 and VP7 first round PCR products were subsequently submitted to the University of Birmingham, School of Biosciences, Functional Genomics, and the Proteomics Facility for Sanger sequencing to determine the genotype. 

### 2.5. Data Analysis

Data were double entered in Epi Info™ version 3.5.3 [36] and analysed using STATA 12.1 (StataCorp LP, 4905 Lakeway Drive, College Station, TX, USA). 

The prevalence of rotavirus infection and rotavirus genotypes in the recruited animals were described. The associations investigated in the univariate analysis using chi square were animal age, sex, animal type, whether the animal was suckling or not, the source of water in the household, the toilet facilities used in the household, and the use of water with soap to wash hands by the investigated humans. 

The animal data were fitted in a mixed effect logistic regression model considering clustering at household and sub-county levels. Factors with a *p*-value ≤ 0.2 and/or biological plausible were considered for the multivariable analysis. The factors in the multivariable analysis were animal age, whether the animal was suckling or not, and animal type. 

The characteristics of the recruited humans from the households were described. The prevalence of the rotavirus and the rotavirus strains found in the humans were also described.

### 2.6. Phylogenetic Analysis

Reference sequences for genome segments encoding the VP7 and VP4 region for RVAs were downloaded from GenBank and visualised using AliView [37]. All of the unverified and duplicate sequences as well as multiple sequences from one country were removed. The reference sequences were aligned with the study sequences using MAFFT [38], and thereafter a maximum-likelihood phylogenetic tree based on the Hasegawa–Kishono–Yano plus empirical base frequencies plus FreeRate model (for VP7) and discrete gamma model (for VP4) were generated using IQ-TREE [39] and run for 1000 pseudo replicates. The trees were visualized using the Interactive Tree Of Life (iTOL) [40] and bootstrap values greater than 75% are shown in the nodes. 

## 3. Results 

### 3.1. Animal Host Characteristics

A total of 1137 domestic animals (418 goats, 281 cattle, and 438 pigs) from 242 households were recruited into this study. Most households reared one or two of the animal types studied: one or two pigs were found in 68/153 (44.4%) households, one or two cattle in 70/108 (64.8%) households, and one or two goats in 56/133 (42.1%) households. 

Here, 693 (60.9%) of the animals were female, and 584 animals (51.4%) were in the age group 0–6 months old. There were 230 animals that were ≤2 months old. Out of these 230 animals, 12 (5.5%) had rotavirus infection. Here, 276 (24.3%) of the animals were still suckling and 117 (10.3%) of the animals had diarrhoea in the two weeks prior to recruitment. Out of the 1011 animals that were asymptomatic, 38 (3.9%) had rotavirus infection (Table 1). 

### 3.2. Risk Factors for Rotavirus Infection in Animals

The frequency of rotavirus infections differed according to the sub-county from which the animals were recruited (Fisher’s exact value = 0.000) (Table 1, Appendix A). There was a difference in the rotavirus prevalence in the investigated domestic animals, where cattle were more likely to be infected with rotavirus than pigs and goats, although the difference was not statistically significant. 

In the multivariable analysis, animals in the age group of 2–6 months, 6–12 months, and 12–18 months were less likely to have rotavirus diarrhoea compared with the age group of 0–2 months old (OR = 0.29, *p* = 0.018, OR = 0.17 *p* = 0.008 and OR = 0.08, *p* = 0.016 respectively) (Table 2). 

The likelihood-ratio test with chi-square = 28.8, *p* < 0.001, showed that the panel-level (sub-county and household) variance components from the random effects model significantly contributed to the total variance and thus to a better model fit for the data.

### 3.3. Human Host Characteristics

A total of 258 human participants aged 2 months–67 years old were recruited from 242 households. Twenty-four of these were aged 2 years or below. Most of the participants were female, namely 138 (53.5%). Of the recruited participants only 31/252 (12.3%) had had diarrhoea in the previous three weeks. Each household was inhabited by an average of eight people (standard deviation = 4). Most of the participants’ source of water was a protected well: 118/258 (45.7%). In 29/248 (11.6%) participants, animals drank from the same source of water. All of the participants had toilets, but only 143/251 (57.0%) washed their hands with soap after toilet use. 

Only two humans (2/258, 0.8%) tested positive for rotavirus. The first sample was collected from a 36-month-old male with a history of diarrhoea in the previous 3 weeks, for a duration of 7 days, and was assigned a P[4] VP4 genotype, whereas its VP7 (G) was non-typeable. Animals drank from this participant’s source of water. This family owned many domestic animals (cattle, pigs, and goats) that tested negative for rotavirus. This participant was reported to play with the animals. The second rotavirus-positive human sample was assigned G9P[8] (GenBank accession no.’s: MG759710, MG759711) and was collected from a 30-month-old female with no history of diarrhoea in the previous three weeks. The household of this participant had pigs and goats that tested negative for rotavirus at the time of sampling. 

## 4. Rotavirus Prevalence and Genotypes in Animals

Forty-five animals were rotavirus-positive. Out of the 60 selected RV ELISA negative samples, only two were rotavirus positive using a more sensitive NSP3 real time RT-PCR test. This brought the total number of investigated animals with RV infection to 47. Thus, the overall prevalence of RV in all of the animals was 4.1% (95% CI: 3.0–5.5%). The prevalence of rotavirus in cattle, using ELISA, was 7.8% (22/281) (95% CI: 6.3–9.6%), while that in goats was 2.4% (10/418) (95% CI: 1.6–3.5%) and that in pigs was 3.4% (15/438) (95% CI: 2.4–4.7%). 

Among the 41 genotyped rotavirus-positive animal samples, six different G-genotypes (G3, G8, G9, G10, G11, and G12) and nine different P-genotypes (P[1], P[4], P[5], P[6], P[7], P[8], P[9], P[10], and P[11]) were observed in different combinations. Mixtures of genotypes were identified in 16/41 (39%) animal samples (Table 3). In 16/41 (39%), no P-genotype was identified and in 11/41(26.8%), no G-genotype was identified, while in 5/41 (12.2%), both the G- and P-genotype could not be determined (Table 3). 

The genotypes found in the pigs were G11 (64.3%), G12 (57.1%), G10 (35.7%), G9 (28.6%), and G8 (7.1%), while two (14.2%) of the samples were G type non-typeable. The P-genotypes found in the pig samples were P[11] (28.6%), P[6] (21.4%), P[4] (14.3%), P[8] (14.3%), and P[10] (14.3%), and two (14.3%) of the samples were P-genotype non-typeable. The combined G- and P- genotypes were G11P[11], G11P[7] and G11P[6]. These were one sample each containing these genotypes and the rest of the samples had either mixed infections or only G- or P-genotype identified.

The G-genotypes found in the goats were G10 (40.0%), G12 (30.0%), G11 (10%) and G9 (10.0%) while 5 samples (50%) were G-genotype non-typeable. The P-genotypes detected in goats were P[6] (20.0%) and P[9] (20.0%), P[7] (10.0%) and P[1] (10.0%) while five samples (50.0%) had P- genotype non-typeable. The goat samples either had mixtures or had one or both G- and P-genotype being non-typeable.

The G-genotypes detected in cattle were G12 (35.3%), G10 (35.3%), G8 (23.5%), G3 (5.9%), and G11(5.3%), while three samples (17.6%) had non-typeable G-genotypes. The P-genotypes in the cattle were P[10] (11.8%), P[6] (11.8%), P[8] (11.8%), P[9] (11.8%), P[11] (11.8%), and P[5] (8.5%), while seven (41.2%) had non-typeable P-genotypes. The combined genotypes detected in five cattle were G3P[10], G8P[6], G8P[9], G10P[8], and G12P[11]. The other samples from cattle had either only G- or P-genotype detected or were mixtures. 

## 5. Human Rotavirus Sequence Results

The VP7 and VP4 sequences of the human sample that were sequenced are closely related to other human rotaviruses that are not of a zoonotic origin (Figure 1 and Figure 2). 

## 6. Discussion

This study sought to determine the prevalence and genotypes of rotavirus across animals and humans found in the same household and the possibility of interspecies transmission of rotavirus. We found a low prevalence of rotavirus infection, namely 0.8% and 4%, in humans and animals, respectively, and a high diversity of rotavirus genotypes. In addition, no interspecies transmission of rotavirus was observed in the households.

The prevalence of rotavirus in the domestic animals in this study was low in comparison with some published reports. This could be because most of the sampled animals in this study were asymptomatic. Another study of healthy pigs in farms in the same area and in northern Uganda also found a low rotavirus prevalence of 0.7% and 0.8%, respectively [41]. A study in Egypt also found a low rotavirus prevalence of 7.9% in symptomatic young goats (kids) [42]. 

However, studies done in Sudan, Tunisia, and Turkey found a high rotavirus prevalence of 21%, 15.4%, and 45%, respectively, in diarrheic and young goats [43,44]. This is as expected, as most rotavirus infections are reported in younger animals [5]. 

While we found a low rotavirus prevalence in pigs, a study in East Africa on asymptomatic pigs in small hold farms found a high rotavirus prevalence of 26.2% [23]. The study in East Africa dealt with large farms raising more than 10 pigs; this may have increased rotavirus transmission among the animals, unlike our study where most households had less than 10 animals. The few animals in the households we sampled may have limited the spread of rotavirus, thus contributing to the low rotavirus prevalence observed. A study in Mozambique also found a low prevalence of rotavirus in the pigs from small holdings [45].

In addition, the study in East Africa used NSP3-specific real time RT-PCR to detect the presence of rotavirus in all of the samples. Although we attempted to increase the sensitivity of detection by retesting 60 specimens using the NSP3-specific real time RT-PCR, it seems unlikely from our data, even using RT-PCR as a screening method, that we may have found a prevalence as high as that reported in the study in East Africa. 

We found a difference in rotavirus prevalence in the different sub-counties. A study in Kenya found a difference in rotavirus prevalence depending on whether the animals were free range or housed [29]. We need to carry out further studies to understand the differences in animal husbandry in the different sub-counties that may account for the difference in the observed rotavirus infection. 

The genotypes detected in pigs were similar to some of the genotypes most commonly found globally, in pigs, namely G3-G5, G9, or G11 in combination with P[6] and P[7] [18,19]. However, the G-genotypes in pigs were different from those reported in a previous study in East Africa, which found G5 and G26 [23]. In addition, in this study, we identified one P[10] strain that is not common in pigs globally [8].

Rotavirus genotypes found in goats were different from those found in studies elsewhere, although reports on goat rotaviruses genotypes are scarce globally. Studies in South Africa found G6P[14], in Korea G3P[3], in Bangladesh G6P[1], and in Turkey G6P[1] and G8P[1] [21,43]. 

The rotavirus genotypes observed in cattle, G10 and G12, were similar to observations in Europe [8] and in Bangladesh, where G10 was prevalent [46]. However, the P-genotypes in our study were more diverse than in a study in Europe and Asia [8,46]. This study also found P[11], similar to reports in a study in Bangladesh, 2009–2010 [46]. Most of the genotypes found in cattle are not those in the available vaccines [11]. 

This study had a high prevalence of mixtures of animal rotavirus genotypes. Some of the mixed genotypes may have been due to cross reactivity/incorrect primer binding due to genetic drift in the rotavirus genomes. However, a high diversity of rotavirus genotypes including mixtures has been reported in various human studies in Africa [47,48,49]. Similarly, animals in this study had a high diversity of rotavirus genotypes including mixtures as the animals and humans stayed in the same environment. A study carried out around the same time in children admitted to four hospitals in central Uganda during 2012/2013 also reported a high prevalence of mixtures of rotavirus genotypes [2]. 

Some of the genotypes detected in the animals, namely G3, G9, G12, P[6], and P[8] had been previously found as mixtures in human samples collected around the same time in Uganda [2]. This emphasizes the possibility of zoonotic and interspecies transmission.

However, the human sample from this study that was sequenced did not show any evidence of interspecies transmission. The samples of the animals in the household were rotavirus negative. Thus, the rotavirus infection in this child was most likely from humans, as evidenced by the phylogenetic analysis. In addition, the second positive human sample that was not sequenced was collected from a household where the animals were rotavirus negative.

## 7. Study Limitations 

Most of the humans studied were asymptomatic and were found to have a low prevalence of rotavirus infection. Thus, we were not able to study factors that were associated with rotavirus infection in the humans in the households. 

The animals in the study were purposively selected, recruiting younger animals and, in addition, the homes in the area were not randomly selected—there was consecutive recruitment of the animals from one home to the next in selected parishes. In addition, the animals were recruited within a short period of two months, thus if there was seasonality of rotavirus infection in animals, the prevalence reported may be under or overreported depending on the season. Although previous studies in humans in Uganda have shown no seasonality of rotavirus infection [2,50].

This study reported a high proportion of non-typeable genotypes. This could be due to the typing tools used and thus recommend designing more primer sets that could be used in the rotavirus genotyping of animals and human samples. The high proportion of non-typeable genotypes observed in this study could have also been due to the poor RNA quality or low viral load that was observed when attempts to sequence the animal samples were carried out as reported elsewhere [51].

## 8. Conclusions and Recommendations

Nevertheless, our study has contributed to the rotavirus genotype data available on bovine, porcine, and caprine hosts in Africa. However, sequence studies of animal samples with mixed genotypes are recommended in order to understand the source of the mixtures, and continued rotavirus surveillance in domestic animals is recommended to determine the relevance for the production and use of rotavirus vaccines. 

The rotavirus prevalence in humans was low and no interspecies transmission of the rotavirus was found.

## Figures and Tables

**Figure 1 viruses-15-00738-f001:**
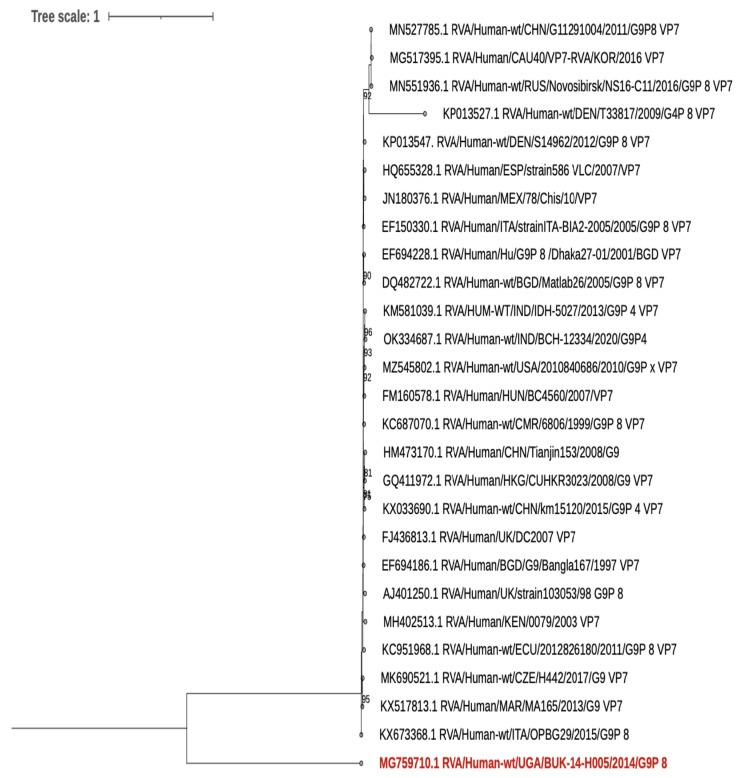
VP7. A Maximum-likelihood phylogenetic tree based on the Hasegawa–Kishono–Yano plus empirical base frequencies plus FreeRate model generated using IQ-TREE (3) and run for 1000 pseudo replicates. The tree was midpoint rooted and bootstrap values greater than 75% are shown at the nodes. The study human strain is in red.

**Figure 2 viruses-15-00738-f002:**
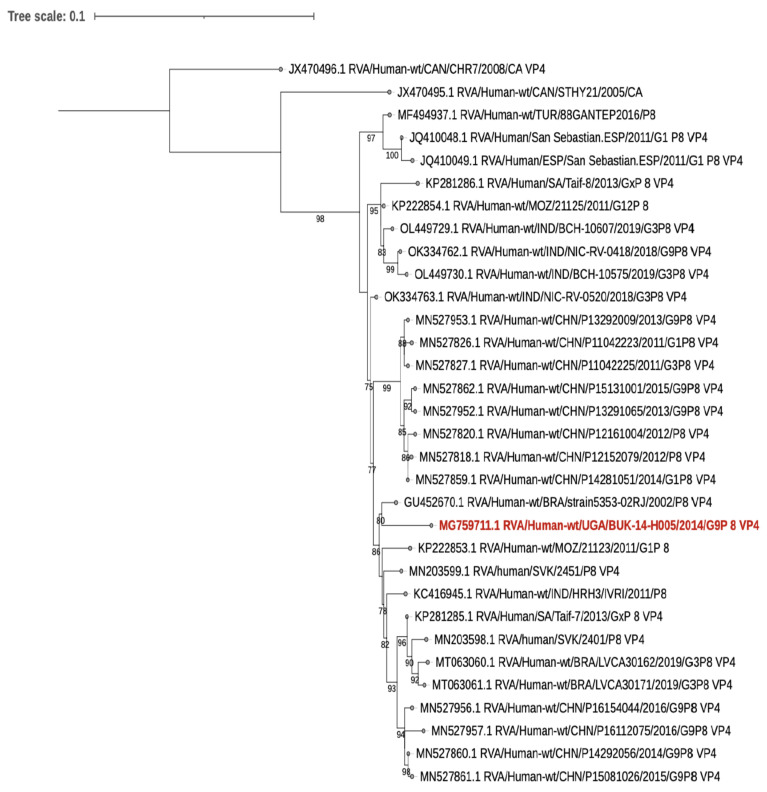
VP4. A Maximum-likelihood phylogenetic tree based on the Hasegawa–Kishono–Yano plus empirical base frequencies plus discrete gamma model generated using IQ-TREE (3) and run for 1000 pseudo replicates. Bootstrap values greater than 75% are shown at the nodes. The study human strain is in red.

**Table 1 viruses-15-00738-t001:** Characteristics of the animals studied for Rotavirus infection in Masaka District, Central Uganda, 2013–2014.

Variable	Rotavirus Negative	Rotavirus Positive (%)	Total N = 1137
**Sex**			
Male	423	21 (5.0)	444
Female	669	24 (3.6)	693
**Age (months) ***			
0–2	218	12 (5.5)	230
2.1–6	338	16 (4.7)	354
6.1–12	205	5 (2.4)	210
12.1–18	99	2 (2.0)	101
18.1–24	76	2 (2.6)	78
≥24.1	133	7 (5.3)	140
**Animal type**			
Pig	423	15 (3.5)	438
Goat	409	9 (2.2)	418
Cattle	260	21 (8.1)	281
**Sub-county**			
Bukakata	174	0 (0.0)	174
Buwunga	88	14 (15.9)	102
Kabonera	178	10 (5.6)	188
Kyanamukaka	195	3 (1.5)	198
Kyesiiga	233	6 (2.6)	239
Mukungwe	231	12 (5.2)	243
**Animal Suckling ***			
No	821	32 (3.9)	853
Yes	265	11 (4.2)	276
**Diarrhoea in past 2 weeks ***			
No	973	38 (3.9)	1011
Yes	110	7 (6.4)	117

* Responses on the variable are less than 1137.

**Table 2 viruses-15-00738-t002:** Univariate and multi-variable analysis of factors independently associated with Rotavirus infection in domestic animals in Masaka District, Central Uganda, 2013–2014.

	Univariate Analysis			Multi-Variable Analysis		
Animal Characteristic	Unadjusted Odds Ratio	95% CI	*p*-Value	Adjusted Odds Ratio	95% CI	*p*-Value
**Sex of Animal**						
Male	1.00					
Female	0.76	0.38–1.51	0.428	-	-	-
**Age of Animal in months**						
0.0–2.0	1.00			1.00		
2.1–6.0	0.39	0.15–1.04	0.059	0.29	0.10–0.81	0.018
6.1–12.0	0.21	0.06–0.74	0.016	0.17	0.04–0.62	0.008
12.1–18.0	0.14	0.02–0.83	0.030	0.08	0.01–0.62	0.016
18.1–24.0	0.26	0.05–1.51	0.134	0.20	0.03–1.37	0.102
>24.0	0.37	0.10–1.30	0.119	0.22	0.05–1.09	0.064
**Animal Type**						
Pigs	1.00			1.00		
Goats	0.57	0.21–1.53	0.266	0.91	0.29–2.80	0.863
Cows	1.05	0.41–2.72	0.915	1.39	0.40–4.83	0.606
**Animal suckling**						
No	1.00			1.00		
Yes	0.90	0.38–2.15	0.816	0.47	0.14–1.53	0.208
**Diarrhoea in past 2 weeks**						
No	1.00					
Yes	1.51	0.54–4.25	0.430	-	-	-

**Table 3 viruses-15-00738-t003:** Rotavirus G- and P-genotypes found in the stool of domestic animals in Masaka District, Central Uganda, 2013 and 2014.

								P-Genotype									
		P[1]	P[6]	P[7]	P[8]	P[9]	P[10]	P[11]	P[5], P[10]	P[6], P[11]	P[6], P[7]	P[7], P[9]	P[8], P[10]	P[9], P[11]	P[4], P[5], P[11]	No P Type identified	Total
	G3						1										**1**
	G8					1										3	**5**
	G10				1											1	**2**
**G-Genotype**	G11		1	1				1									**3**
	G12							1								1	**2**
	G9,G11															1	**1**
	G10,G12		1			2			1			1				4	**9**
	G11,G12										1				2		**3**
	G8,G9, G10,G12				1		1										**2**
	G9,G10, G11,G12													1		1	**2**
	No G-Type identified	1	1		1			1		1			1			5	**11**
	**Total**	**1**	**4**	**1**	**3**	**3**	**2**	**3**	**1**	**1**	**1**	**1**	**1**	**1**	**2**	**16**	**41**

## Data Availability

The data presented in this study are available in the links below: https://uvri.go.ug/sites/default/files/animal%20%20type%20and%20RV%20genotypes_17th%20Feb%202023.xlsx; https://uvri.go.ug/file/841#overlay-context=file/839. Accessed on 20 February 2023. The GenBank accession numbers for the sequences are MG759710 and MG759711.

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
