# Peer review of "Co-Surveillance of Rotaviruses in Humans and Domestic Animals in Central Uganda Reveals Circulation of Wide Genotype Diversity in the Animals"

_viruses, 2023, doi:10.3390/v15030738_

Round 1
Reviewer 1 Report
This manuscript presents the results of a sizeable survey of rotaviruses circulating in humans and livestock in Uganda. Rotavirus surveillance in this region is limited, and hence these findings are valuable for understanding overall importance of this virus. Interestingly, virus prevalence in humans in this study was found to be very low, with no evidence of any zoonotic transmission. A wide range of rotavirus genotypes were detected in livestock. Overall this is a useful and succinct study, that is clearly written with sound conclusions drawn. Methods used are well described and used appropriately.
My only major comment is regarding the dates when the samples were collected - this is now almost a decade ago. Is it possible to explain the delay in exploring / publishing these findings? Although I still think the results are interesting, there is concern that these may not reflect the current situation almost a decade later.
Minor comments are as follows:
Abstract line 41: ‘Animals aged 2 to 18 months were less likely to have rotavirus infection’ needs clarifying. Can you add a sentence to clarify which age group they are being compared to, i.e. animals <2 months?
Introduction
- Reference 1 – suggest finding a newer reference if possible, this is now >6 years old
- There is a good discussion on vaccines available worldwide, but it would be good to provide detail about which vaccines were used in Uganda at the time of sample collection. It would also be useful to give the latest vaccine situation in the country.
Methods
- Line 95 – it would be useful to quickly mention the infection prevalence use in calculations rather than referring to a reference.
- Would it be possible to include a map of sampling region/sites?
Results
- Table legends – please check the placement of the legends, the first table in the manuscript doesn’t have a legend directly associated, but I think it is meant to be lines 186-188. I cannot tell what the second table is between lines 188-189. Is there a title and another way to present this information?
- Line 171 – ‘one-to-two’ is a slightly confusing way of describing the numbers of animals involved. One or two? Up to two?
- Table 2 – this is a great table. Is it possible to colour code to show which species each of the different genotypes were found in?
Reviewer 2 Report
The manuscript by Bwogi and colleagues describes a cross-sectional study conducted in Uganda to determine rotavirus (RV) prevalence and genotypes in animals and humans living in the same household. Despite the availability of multiple vaccines, RV-induced gastroenteritis still represents a significant health burden in many parts of the world. As RVs are capable of both zoonotic transmission and reassortment, monitoring circulating animal and human RV strains remains crucial in order to understand the emergence of novel or unusual genotypes as well as the effect of vaccination.
The manuscript is interesting, concise and well written. It is unfortunate that the number of RV-positive human samples was too low for a more in depth analysis. However, the authors clearly state this limitation (and others) in their manuscript. The manuscript sheds some light into the RVA prevalence in animals in Uganda and the data could help to design improved studies.
Specific comments:
1. Line 32: In the abstract, it is stated that RV infection in humans was detected using ELISA, but in the manuscript it is written that real-time RT-PCR was used for human samples and a subset of animal samples. Please correct.
2. Line 33: “nested RT-PCR” instead of “nested real-time PCR”
3. Line 33 and throughout: “G type/P type/G and P type/G and P genotype etc.” use hyphens or no hyphens consistently
4. Line 34: “a non-typeable strain” instead of “the non typable strain”
5. Line 41: add “in comparison to animals below 2 months of age”
6. Line 50: Consider adding “Cohen et al: Aetiology and incidence of diarrhoea requiring hospitalisation in children under 5 years of age in 28 low-income and middle-income countries: findings from the Global Pediatric Diarrhea Surveillance network. BMJ Global Health 2022,” who estimate that the number of RV-attributable deaths is higher than previously estimated.
7. Line 68: After listing the licensed vaccines, consider adding if RVA vaccination in animals/humans occurs or is planned in Uganda.
8. Line 89: “study” instead of “studied”
9. Lines 93-95: Consider adding that the published prevalences are based on cattle under two months of age and diarrheic piglets aged from 1 to 43 days. Also indicate precision and confidence intervals for completeness.
10. Lines 102-112: As animals and humans of all age groups with and without diarrhea were enrolled in this study, overall prevalence of RVA infection was low. While this was discussed in detail for animals, it would be helpful to shortly discuss prevalence in humans of different age groups with/without diarrhea
11. Lines 114-116: Did data from recruited humans include history of diarrhea in the previous two weeks or three weeks?
12. Lines 141-142: Did the authors mean NSP3-specific real-time RT-PCR or was it really a VP6-specific one?
13. Lines 174-175: How many animals under 2 months had diarrhea and tested positive for RVA? Where there many asymptomatic infections? A more detailed report of actual numbers could be helpful here.
14. Line 177: Consider adding a map showing the district, counties, sub-counties where samples were collected, ideally together with the number of positive samples.
15. Table 1: There seems to be a formatting problem with the tables and description. Please fix.
16. Line 206: Did any of the two households report diarrhea in animals?
17. Fig 1 and 2: The figures could be enlarged. The bootstrap values are barely visible and should be displayed in a larger font.
18. Discussion beginning: It would be helpful to start with a short paragraph summarizing the findings.
19. Discussion line 20: Consider changing “in comparison to the published reports” to “in comparison to some published reports”.
20. Discussion line 21: How many asymptomatic animals were RVA-positive? How common is asymptomatic RVA infection in the examined animals.
21. Discussion line 23: Consider adding “Boene et al.: Prevalence and genome characterization of porcine rotavirus A in southern Mozambique. Infection, Genetics and Evolution 2021,” who also showed low prevalence in smallholdings.
22. Discussion line 51: Is there data on how animal RVA vaccines affect circulating RVAs in animals?
23. Discussion line 58: A large proportion of genotypes was non-typeable in this study. Could this indicate that better typing tools are necessary to identify unknown genotypes? Do the authors have an indication that other factors, such as RNA quality or a low viral load were an issue? Consider adding this to study limitations.
24. Discussion lines 73-75: Consider adding a statement about the seasonality of RVA infections in humans in East Africa.
